

# Effects of different soil types on gas exchange parameters and fruits of *Hippophae rhamnoides* ssp. *mongolica* "Ulanshalin" plants

Fanjing Bu[1], Yuefeng Guo[1] and Wei Qi[2]

[1] College of Desert Control Science and Engineering, Inner Mongolia Agricultural University, Hohhot, Inner Mongolia, China
[2] Inner Mongolia Autonomous Region Water Conservancy Development Center,, Hohhot, Inner Mongolia, China

## ABSTRACT

**Background**. This study aims to explore the growth and production potential of *Hippophae rhamnoides* ssp. *mongolica* "Ulanshalin," a pioneer species of soil and water conservation, after being planted in the Yellow River Basin area with serious soil erosion. An analysis of the differences in photosynthesis and fruit yield of *H. rhamnoides* plants grown in two typical soils in the watershed is key to understanding whether local conditions are suitable for the growth and yield of *H. rhamnoides,* as well as the impact of the plants on soil and water conservation.

**Methods**. During the growing season, diurnal changes in the gas exchange parameters of *Hippophae rhamnoides*-like plants growing in Loess soil and Aeolian soil were continuously monitored, and the effects of total nitrogen (TN) and other elements on the net photosynthetic rate ($P_N$) of the plants were analyzed and compared in the two different soil types. The morphological and quality differences of *Hippophae rhamnoides* fruits were also compared after reaching the ripening stage.

**Results**. (1) There was a significant difference in the composition of Loess soil and of Aeolian soil. The organic matter content and AK content of the Loess soil was significantly higher than in the Aeolian soil, and the pH was closer to neutral. However, the TK content, TP content, and AP content of the Aeolian soil were slightly higher than in the Loess soil, the pH was higher, and it was alkaline. (2) After controlling the light and temperature, with all other external factors consistent, the daily variation trend of $P_N$, $T_r$, and $G_s$ in the leaves of *H. rhamnoides* plants growing in the two different soils were basically the same. There were differences, however, in when these factors reached their peaks. Soil composition had an impact on the photosynthetic characteristics of *H. rhamnoides*, with TN, TP, AP, and SOM being the main factors promoting the photosynthetic rate of *H. rhamnoides* $P_N$. The peaks of $P_N$, $T_r$, and $G_s$ of *H. rhamnoides* plants growing in Aeolian soil were higher than those growing in Loess soil. (3) The average stem length of *H. rhamnoides* plants growing in Aeolian soil was higher than the plants growing in Loess soil. The number of thorns in the branches of plants in the Aeolian soil was relatively low, and the weight of 100 fruits (28.28 g) was significantly higher than the weight of 100 fruits of the *H. rhamnoides* plants grown in Loess soil (11.14 g).

**Conclusions**. The results of this study show that in the Yellow River Basin area, Aeolian soil is more conducive to the growth of *H. rhamnoides* plants than Loess soil. *H.*

Corresponding author
Yuefeng Guo, bfj@emails.imau.edu.cn

*rhamnoides* plants growing in Aeolian soil had good adaptability and stress resistance, and a larger potential for fruit production. These findings provide insights for ecological restoration and the creation of economic value in the Yellow River Basin area.

## INTRODUCTION

The Inner Mongolia section of the Yellow River Basin is an area of China with serious soil erosion, and the main soil types in the basin are Loess soil and Aeolian soil (*Tong et al., 2015*; *Ge, Li & Peng, 2019*). The natural environment of the basin is relatively harsh, with little annual rainfall, leading to significant water shortage, difficulty in plant rooting, and poor tree growth. These conditions seriously reduce the survival rate of plants, and a significant amount of native plants in the area have died, affecting the sustainable development of forest stands (*Bu et al., 2022*). It is, therefore, necessary to explore the growth and production potential of other plants with soil and water conservation functions in this area to improve the local ecological environment and improve the economic output of the area.

*Hippophae rhamnoides* ssp. *Mongolica,* or "Ulanshalin," is a subspecies of *Hippophae rhamnoides L. subsp. Mongolica Rousi*, a hybrid created by experts in Mongolia through cross-breeding. These plants have large fruits with a thick peel and are known for their cold resistance and strong adaptability to the environment. *Kong (2020)* conducted a two-year introduction and planting test on different varieties of *H. rhamnoides* on the eastern Loess Plateau, and found that when comparing survival rate, plant height, crown width, and base diameter, "Ulanshalin" performed relatively well, and other varieties of *H. rhamnoides* performed poorly. We previously attempted to introduce and cultivate *H. rhamnoides* on a large scale in the Yellow River Basin area, but the growth and production of the plants differed significantly due to the different soil types in the study area. Because of this, we were unable to predict the type of soil suitable for the growth of "Ulanshalin" in the region. This led to this analysis of soil types.

Soil is the basis for plant survival. Different soil types have different nutrient content, forming different soil environments. These differences affect the exchange of water, air, and nutrients between soil and plants, ultimately affecting plant growth and yield (*Bauhus, Paré & Cté, 1998*; *Bechtold & Naiman, 2006*; *Berg & Smalla, 2009*). Photosynthesis plays an important role in plant growth and development, as well as the regional carbon cycle. It is the most important physiological process, and is affected by ecological factors such as light, temperature, water, and $CO_2$ concentration, so more attention should be paid to the impact of soil nutrients on photosynthesis (*Merlo, 2022*; *Adamczyk Szabela & Wolf, 2022*). *He et al. (2022)* found that *Rumex nepalensis* plants had different photosynthetic rates when planted in grassland soil, shrub soil, and forest soil, with plants in grassland soil having the largest photosynthetic rates followed by shrub soil and then forest soil. *Jin, Zheng & Liu (2021)* found that higher potassium levels led to a higher net photosynthetic rate in cotton,
showing that soil elements impact plant photosynthesis. Based on this understanding, we studied the impact of soil elements on photosynthesis in *H. rhamnoides* plants.

The Yellow River Basin is a hotspot for *H. Rhamnoides* research. Current research on the photosynthetic characteristics of *H. rhamnoides* and its influencing factors in China mostly focuses on environmental factors such as soil moisture and temperature. *Xia et al. (2015)* found that when the relative water content (RWC) of soil was 50.4%, the $P_N$ of *H. rhamnoides* remained at a high level throughout the day, and there was no "lunch break" phenomenon. When RWC exceeded this range, the $P_N$ of the whole day decreased, accompanied by the photosynthetic "lunch break" phenomenon. When the RWC decreased to 36.3%, the photosynthetic "lunch break" phenomenon no longer occurred. *Jin et al. (2011)* found that photosynthetically active radiation (PAR) and air relative humidity (RH) are the dominant environmental factors controlling changes in *H. rhamnoides* $P_N$. These scholars also studied the influence of the natural environment on the gas exchange parameters of *H. rhamnoides* plants in the Yellow River Basin, but differences in the soil parameters made the results unclear.

In this study, we measured differences in gas exchange parameters and fruit yield of *H. rhamnoides* plants in two typical soils in the Yellow River Basin area, accounting for the possible effects of external factors such as light and temperature, in order to clarify the impact of soil elements on $P_N$. We examined fruit shape and yield to provide a theoretical basis for the cultivation and promotion of *H. rhamnoides* in the Yellow River Basin. We tested the following hypotheses: (1) the gas exchange parameters of *H. rhamnoides* would differ significantly between the two soil types, and $P_N$ would be affected by soil factors, and (2) Aeolian soil would be more conducive to the growth and fruiting of *H. rhamnoides* plants than Loess soil.

# MATERIALS & METHODS

## The study area

The study area was located in the Inner Mongolia Autonomous Region of China (39°42′–39°50′N, 110°25′–110°48′E; Fig. 1; *Wang et al., 2022*). The Inner Mongolia section of the Yellow River Basin is an area with uneven terrain, many ravines, are serious soil erosion. The main soil types in the basin are Loess and Aeolian. The Yellow River Basin has a semi-arid monsoon climate, an average altitude of about 800–1,590 m, with sunshine hours between 2900–3100 h, and an annual frostless season of 148 d. The area has an average annual precipitation of 400 mm, which is densely distributed in July–August. The annual average evaporation is 2,093 mm, the annual average temperature is 6.2−8.7 °C, and the annual cumulative temperature ≥10 °C is 2,900∼3,500 °C. The vegetation in the area is primarily small shrubs and grasses such as *Caragana*, *Pinus,* and *Medicago*, with *Artemisia* and *Phragmites* also quite common in the area.

## Experimental design
### Methods

This experiment was carried out on June 16, 2022 in the Soil and Water Conservation Science and Technology Demonstration Park in the Yellow River Basin. *H. rhamnoides*

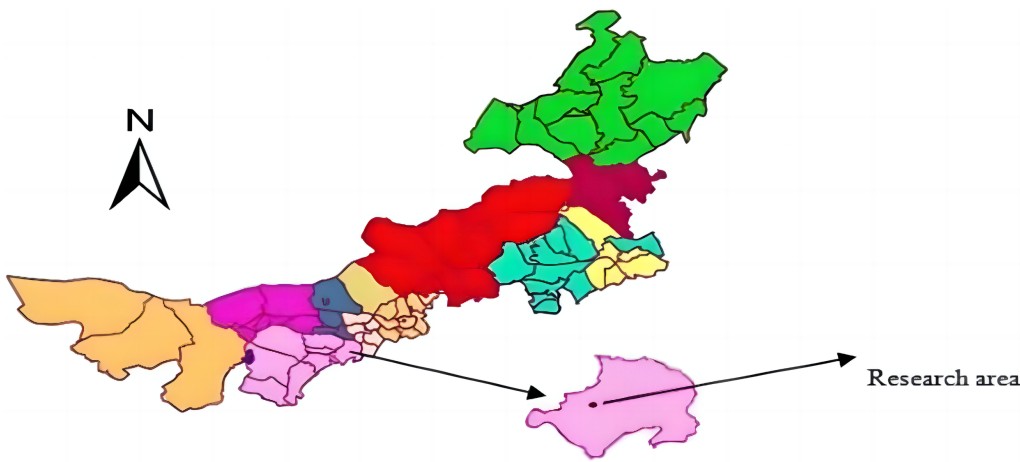

**Figure 1  The location of the study area.** The study area was located in the Inner Mongolia Autonomous Region of China (39°42′–39°50′N, 110°25′–110°48′E). Figure source credit: 10.27229/d.cnki.gnmnu.2020. 000695.

plants of the same age, management, and protection measures were selected as the research object. The planting density was 3× 3 m, meaning there was 3 m between the rows of *H. rhamnoides* plants and 3 m between the individual plants in the rows. In the park, two 50 m × 50 m sample plots were set up: one with pure Loess soil (Sample Plot 1) and the other with pure Aeolian soil (Sample Plot 2). Each wood scale of *H. rhamnoides* in the sample plot was carried out. Nine *H. rhamnoides* plants with straight trunks, good growth, and no diseases or pests were randomly selected from each plot as the sample plants for investigation and sampling. In order to increase the reliability of the experiment, the standard cluster of each plot was averaged. The basic information of the sample plots is listed in Table 1.

### Detection of soil physiochemical properties

On the north side of the 18 clusters of *H. rhamnoides* plants selected from the two sample plots, soil columns were collected 10 cm from the base of the plant at the 30 cm layer using a soil auger. Three bags of soil columns were taken from each plant, for a total of 54 bags, which were then sealed, labeled, and brought back to the laboratory for analysis. The soil samples were then air-dried, depurated, ground, and screened at 0.15 mm and two mm to determine the total potassium (TK, g/kg), total phosphorus (TP, g/kg), total nitrogen (TN, g/kg), and other nutrient content of the soil. The soil potential of hydrogen (pH) was determined using the potentiometric method. Available phosphorus (AP, mg/kg) was determined using the sulfuric acid extraction-molybdenum blue colorimetric method. Available potassium (AK, mg/kg) content was measured using an ammonium acetate-flame photometer. Soil organic carbon content was determined by potassium dichromate oxidation. Soil organic carbon content was then multiplied by a conversion coefficient of 1.724 to obtain the soil organic matter (SOM, g/kg) content. TP content was determined using the sulfuric acid-perchloric acid digestion method, the TK content was determined

**Table 1  Vegetation overview of the sample plots.**

| Sample | Forest age/year | Base diameter/cm | Plant height/cm | Crown width EW/cm | Crown width SN/cm | The main herbaceous plants under the forest |
|---|---|---|---|---|---|---|
| Sample 1 | 5a | 2.40 ± 0.10 | 112.67 ± 9.50 | 104.33 ± 11.67 | 111.67 ± 10.26 | *Artemisia argyi Levl* *ArtemisiacapillarisThunb* *Phragmites australis* |
| Sample 2 | 5a | 3.33 ± 0.46 | 262.67 ± 23.69 | 196.67 ± 48.56 | 197.33 ± 35.85 | *Alopecurus pratensis* *Cyperus rotundus* *Artemisia argyi Levl* *Phragmites australis* |

**Notes.**

EW for the crown of the east–west direction; SN is the crown of the north-south direction.

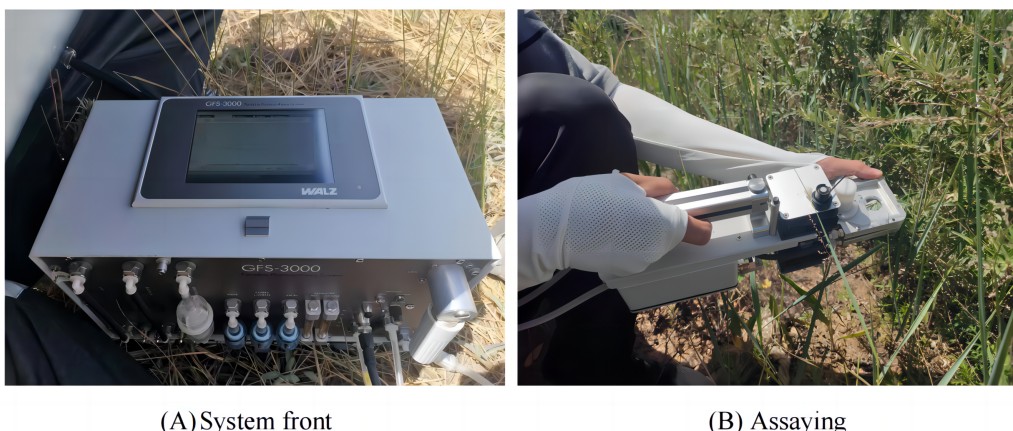

(A) System front          (B) Assaying

**Figure 2  Utilization of GFS-3000.** Sea buckthorn gas exchange parameters are being measured using an instrument.

using sodium hydroxide iodine dissolution-flame photometry, and the TN content was determined using the semi-micro Kelvin method (*Lin, 2004*).

### Determination of gas exchange parameters

The trial was conducted on a clear day in late June 2022. Photosynthetic capacity under natural conditions was measured with a portable, open, fully-controlled flow cytometric gas exchange fluorescence measurement system (GFS-3000, Walz, Germany; Fig. 2A; *Laine, Korrensalo & Tuittila, 2022*).

The system has high accuracy in determining plant gas exchange parameters. It also has a comprehensive measurement design, and the $P_N$ ($P_N$, umol m$^{-2}$ s$^{-1}$), transpiration rate ($T_r$, mmol m$^{-2}$ s$^{-1}$), stomatal conductance ($G_s$, mol$^{-2}$ s$^{-1}$), and intercellular $CO_2$ concentration ($C_i$, umol $CO_2$ mol$^{-2}$) of plant leaves can be measured synchronously after placing the leaves in the leaf chamber. The advantage of this instrument is that it does not ignore the photosynthesis measurement on the back of the leaf, does not cause damage to the leaf, and is suitable for repeat measurements. Before measurement, the leaves were placed in the leaf chamber for 5 min to gain photosynthetic stability (*He et al.,*

*2022*). During the measurement process, it was verified that all leaves were fully in the leaf chamber, and the leaf chamber was not leaking air (Fig. 2B). When the relative value of the measurement index was stable or fluctuated slightly, a set of data was recorded and stored, with the process repeating after replacing the leaf in the leaf chamber. Three leaves from each *H. rhamnoides* plant were randomly selected, with each leaf being measured every two hours, for a total of three times, between the hours of 7:00–19:00. The average of nine times was taken, and it was determined continuously for one week (*Chen et al., 2023*).

### Determination of fruit indicators

After the fruits were ripe, 100 fruits were picked from each sample *H. rhamnoides* plant and weighed together in an electronic balance to calculate the cumulative weight of 100 fruits. The measurement results of the weight of 100 fruits from each of the nine sample plants from each plot were then averaged. A total of 50 fruits were randomly selected by the shape of the fruit, and the longitudinal diameter and transverse diameter were determined using vernier calipers. These fruits were then divided according to the longitudinal to transverse ratio: flat fruit had a longitudinal diameter ratio less than 0.9, round fruit had a longitudinal diameter ratio 0.91~1.10, oval-shaped fruit had a longitudinal diameter ratio 1.11~1.40, and cylindrical fruit had a longitudinal diameter ratio greater than 1.41 (*Chen, Liu & Luo, 2014*).

### Data processing and analysis

One-way analysis of variance (ANOVA) was used to test for differences in the content of soil elements between Loess and Aeolian soil, and the least significant difference method (LSD; significant at $p < 0.05$ level) was used for post-hoc multiple comparisons. In order to accurately analyze whether soil element content affects the $P_N$ process of *H. rhamnoides*, SPSS (ver. 22.0; SPSS Inc., Chicago, IL, USA) was used for a pathway analysis (*Liao et al., 2021*). Plotting was finished in Origin 2018.

## RESULTS

### Soil types and their physical and chemical properties

Table 2 shows that the chemical composition of Loess soil varied greatly. AK content was the highest, accounting for 69% of the measured sample, followed by TK at 13%. SOM and AP content were both low, accounting for 6% and 4% of the total soil sample, respectively. TN and TP content were the lowest, with both measuring below 0.04% of the soil sample, and the pH value was 8.6~8.68, which is weakly alkaline. The chemical composition of Aeolian soil also varied greatly. AK content was the highest, accounting for 62% of the sample, followed by TK at 16%. SOM and AP content were both low, accounting for 3% and 8% of the total soil sample, respectively. TN and TP content were the lowest, with both measuring below 0.04% of the soil sample, and the pH value was between 9.22~9.3, with alkaline reaction. There was a significant difference in the composition of Loess soil and Aeolian soil. The organic matter content and AK content of Loess soil were significantly higher than in Aeolian soil ($P < 0.05$), and the pH of the Loess soil was closer to neutral. However, the TK content, TP content, and AP content of the Aeolian soil were slightly

**Table 2 The physicochemical properties of Loess and Aeolian soils.**

| Soil Type | TK (g/kg) | TP (g/kg) | TN (g/kg) | SOM (g/kg) | AP (mg/kg) | pH | AK (mg/kg) |
|---|---|---|---|---|---|---|---|
| Loessial soil | $17.78 \pm 1.22a$ | $0.28 \pm 0.04b$ | $0.50 \pm 0.08bc$ | $9.41 \pm 2.04bc$ | $6.00 \pm 2.28bc$ | $8.64 \pm 0.04c$ | $96.73 \pm 12.61c$ |
| Aeolian soil | $19.17 \pm 0.60a$ | $0.38 \pm 0.03b$ | $0.40 \pm 0.20c$ | $3.63 \pm 1.33c$ | $9.76 \pm 0.48d$ | $9.26 \pm 0.04d$ | $79.25 \pm 3.21d$ |

**Notes.**

Data are all expressed as mean ± standard deviation; different lowercase letters indicate significant differences among treatments ($P < 0.05$).

higher than the Loess soil ($P < 0.05$), the pH was higher, and it was alkaline. TN content did not differ significantly between the two soils.

## Diurnal variation of gas exchange parameters

Photosynthesis plays a decisive role in the growth and development of green plants, and the size of the gas exchange parameters can reflect the strength of plant growth (*Gobu et al., 2022*). The diurnal variation in four common gas exchange parameters ($P_N$, $T_r$, $C_i$, $G_s$) were compared under different soil types to continue to analyze the possible influence of soil chemical composition on gas exchange parameters (*Danish et al., 2022*).

Figure 3A shows that the daily variation of *H. rhamnoides* $P_N$ under the two soil conditions followed a bimodal curve, with both showing the "photosynthetic" lunch break, phenomenon, and the change law was roughly similar. The $P_N$ of the Loess soil reached the first peak at 10:00 (7.19 umol m$^{-2}$ s$^{-1}$), the second peak at 14:00 (5.45 umol m$^{-2}$ s$^{-1}$), with the trough at 12:00. The $P_N$ of the Aeolian soil reached the maximum peak at 11:00 (11.14 umol m$^{-2}$ s$^{-1}$), with a trough around 12:00, and a small peak at 14:00 (10.04 uumol m$^{-2}$ s$^{-1}$). *H. rhamnoides* $P_N$ differed by soil type at various times of the day, but both soils followed the same trend of a large peak value with relatively rapid growth before reaching the peak, with a slightly slower decline after the peak. The $P_N$ in Aeolian soil was higher than in Loess soil, and the physiological activity intensity was greater in the Aeolian soil.

Figure 3B shows that the diurnal variation of *H. rhamnoides* $T_r$ also differed by soil type. The $T_r$ of *H. rhamnoides* in Loess soil followed a bimodal curve, with the first peak of 2.80 mmol m$^{-2}$ s$^{-1}$ at about 10:30, a trough at 14:00, and a second peak of 1.96 mmol m$^{-2}$ s$^{-1}$ at about 14:00. The $T_r$ of *H. rhamnoides* in Aeolian soil showed a unimodal change, with a maximum value of 4.72 mmol m$^{-2}$ s$^{-1}$ at 13:00, and then $T_r$ continued to decrease until it was at its lowest measured level at 18:00. *H. rhamnoides* $T_r$ peaked in Aeolian soil 2.5 h later than the first peak in the Loess soil. *H. rhamnoides* $T_r$ in the Aeoloan soil was slightly higher than in the Loess soil.

Figure 3C shows that the diurnal variation of *H. rhamnoides* $C_i$ was more complex and differed between the two soil types. *H. rhamnoides* $C_i$ in Loess soil followed an inverted parabola that first fell and then rose, starting at 360.04 umolCO$_2$ mol$^{-2}$ at 08:00, decreasing to the lowest measured value of 271.66 umolCO$_2$ mol$^{-2}$ at 12:00, and then rising again. *H. rhamnoides* $C_i$ in Aeolian soil followed a bimodal curve, reaching a maximum peak of 346.86 umolCO$_2$ mol$^{-2}$ at 9:00, decreased sharply to a minimum value of 274.96 umolCO$_2$ mol$^{-2}$ at 12:00, rising again to the second peak of 339.60 umolCO$_2$ mol$^{-2}$ at 16:00, and

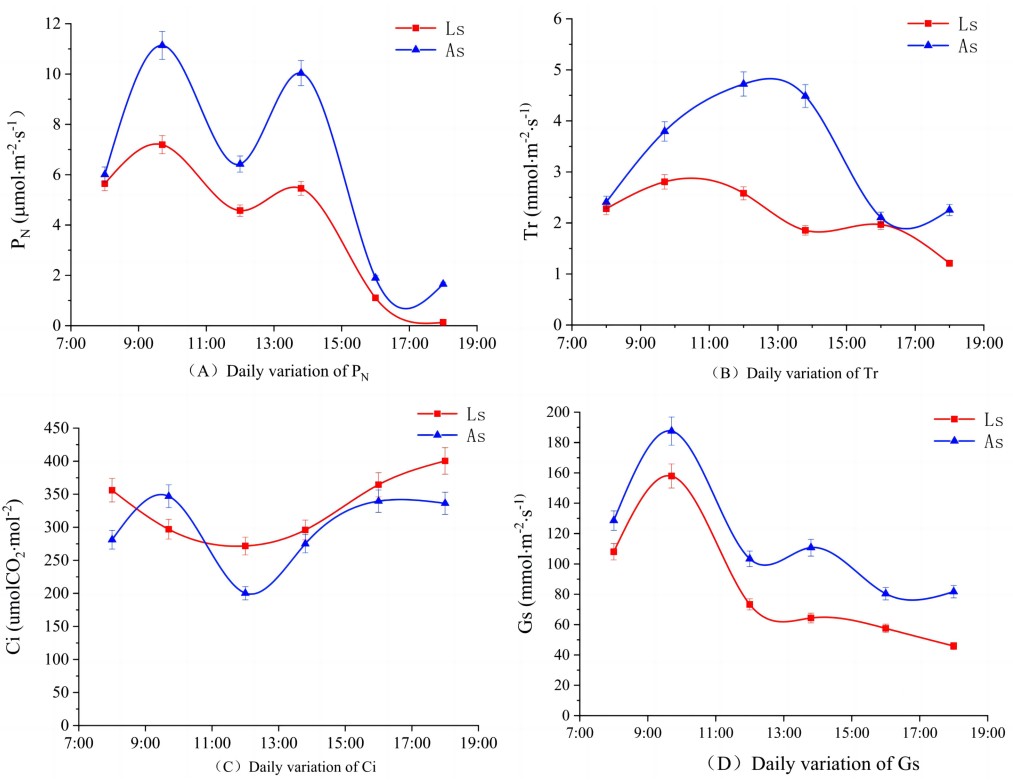

**Figure 3** **Diurnal variation in gas exchange parameters by soil type.** Ls, Loess soil; As, Aeolian soil. Diurnal curve plot of sea buckthorn gas exchange parameters.

then slowly decreasing again. $C_i$ in Loess soil reached its first peak about 1 h before Aeolian soil.

As shown in Fig. 3D, the daily variation of *H. rhamnoides* $G_s$ did not differ significantly in the two soils. Both followed a bimodal curve with peak values appearing at 10:00 followed by a trough at 12:00, another small peak at 14:00, and then gradually decreasing. The two *H. rhamnoides* $G_s$ peaks in Loess soil were 157.95 mol$^{-2}$ s$^{-1}$ and 64.34 mol$^{-2}$ s$^{-1}$, and the two *H. rhamnoides* $G_s$ peaks in Aeolian soil were 187.56 mol$^{-2}$ s$^{-1}$ and 110.72 mol$^{-2}$ s$^{-1}$.

## Path analysis of net photosynthetic rate and soil factors

The relationship between soil and photosynthesis is relatively complex because all soil factors affect plant photosynthesis to varying degrees. A pathway analysis was performed in order to analyze the impacts of TN, TK, TP, and other soil elements on *H. rhamnoides* $P_N$. The analysis results are shown in Fig. 4 and in Tables 3 and 4. The pathway analysis of soil factors and *H. rhamnoides* $P_N$ showed that TN, AP, PH, TK, and $P_N$ were correlated, TN and AP were positively correlated with $P_N$, with correlation coefficients of 0.552 and 0.289, respectively, and pH and AK were negatively correlated with $P_N$. TK, TP, and SOM were not correlated with *H. rhamnoides* $P_N$ (Table 3, Fig. 4A). TN maximally promoted *H. rhamnoides* $P_N$ (1.055), and the direct effect was significantly higher than that of other factors, with AP having the second largest impact on $P_N$ (0.591). The direct effect of

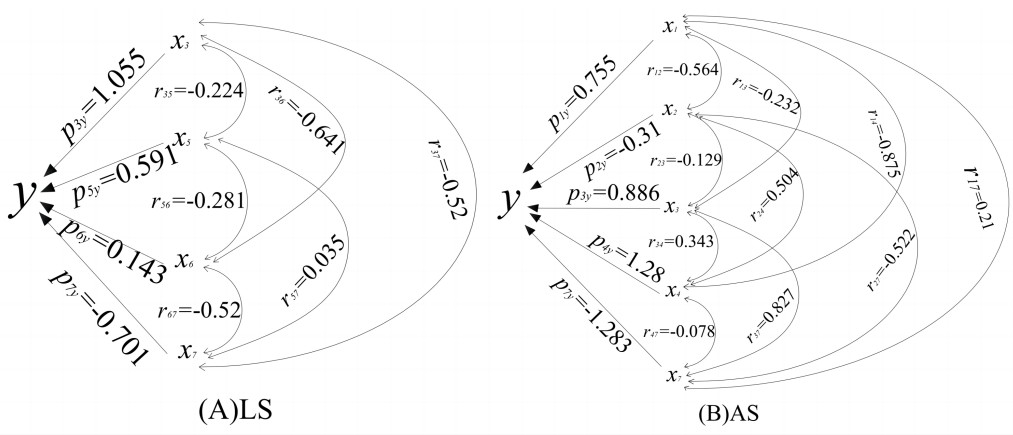

**Figure 4** **Pathway coeûcient of PN and soil factor of H. rhamnoides.** LS: Loess soil; AS: Aeolian soil. Ae-olian soil and loess soil were analyzed with sea buckthorn PN diameter analysis coefficients, respectively.

**Table 3** **Pathway analysis of Loess soil factors and *H. rhamnoides* $P_N$.** Pathway analysis of loess soil factors and rhamnose PN, direct correlation coefficients and indirect coefficients.

| Soil factors | Correlation coefficient | Direct path coefficient | Indirect connection diameter coefficient | | | |
|---|---|---|---|---|---|---|
| | | | TN | AP | pH | AK |
| TN | 0.552 | 1.055 | – | −0.132 | −0.092 | −0.279 |
| AP | 0.289 | 0.591 | −0.236 | – | −0.040 | −0.025 |
| pH | −0.334 | 0.143 | −0.676 | −0.166 | – | 0.365 |
| AK | −0.335 | −0.701 | 0.420 | 0.021 | −0.074 | – |

**Table 4** **Pathway analysis of Aeolian soil factors and *H. rhamnoides* $P_N$.** Pathway analysis of aeolian soil factors and rhamnose PN, direct correlation coefficient and indirect coefficient.

| Soil factors | Correlation coefficient | Direct path coefficient | Indirect connection diameter coefficient | | | | |
|---|---|---|---|---|---|---|---|
| | | | TK | TP | TN | SOM | AK |
| TK | −0.665 | 0.755 | – | 0.175 | −0.206 | −1.120 | −0.269 |
| TP | 0.465 | −0.31 | −0.426 | – | −0.114 | 0.645 | 0.670 |
| TN | 0.128 | 0.886 | −0.175 | 0.040 | – | 0.439 | −1.061 |
| SOM | 0.867 | 1.28 | −0.661 | −0.156 | 0.304 | – | 0.100 |
| AK | −0.33 | −1.283 | 0.159 | 0.162 | 0.733 | −0.100 | – |

AK content was negative (−0.701), indicating that AK content alone does not have a significant effect on *H. rhamnoides* $P_N$, but AK indirectly promotes $P_N$ by affecting TN content ($P_{AK \times TN} = 0.420$, $P_{AK \times TN}$ represents the indirect diameter coefficient of SOM to AP). A stepwise regression analysis was used to further determine the influence of soil factors on *H. rhamnoides* $P_N$, and the following regression equation was obtained:

$$y = -0.577 + 2.649x_3 + 0.057x_5 + 0.594x_6 - 0.008x_7, (x_3 = TN, x_5 = AP, x_6 = pH, x_7 = TK, P < 0.05).$$

The equation results show that TN is the most important soil factor in promoting *H. rhamnoides* $P_N$. AP and TK content also have an important effect on *H. rhamnoides* $P_N$.

The diameter analysis of soil factors and *H. rhamnoides* $P_N$ in Aeolian soil showed (Table 4 and Fig. 4B) that SOM, TP, and TN were positively correlated with $P_N$, with correlation coefficients of 0.867, 0.465, and 0.128, respectively, and TK, AK, and AP were not correlated with *H. rhamnoides* $P_N$.

In Aeolian soil, SOM maximized *H. rhamnoides* $P_N$ (1.28), and the direct effect was significantly higher than other factors, followed by TN (0.886) and TK (0.755). The direct effect of TP content was negative ($-0.31$), indicating that TP content alone did not have a significant effect on *H. rhamnoides* $P_N$, but it indirectly promotes $P_N$ by affecting SOM content ($P_{TP \times SOM} = 0.645$, $P_{TP \times SOM}$ indicates the indirect diameter coefficient of TP to SOM). The direct effect of AK content was also negative ($-1.283$), but it indirectly promotes $P_N$ ($P_{AK \times TN} = 0.733$) by impacting TN content. In order to further determine the influence of soil factors on *H. rhamnoides* $P_N$, a stepwise regression analysis was used to obtain the regression equation: $y = 12.228 + 1.936x_1 - 38.079x_2 + 32.832x_3 + 5.849x_4 - 0.828x_7$, ($x_1 = TK$, $x_2 = TP$, $x_3 = TN$, $x_4 = SOM$, $x_7 = AK$, $P < 0.05$). The equation results show that the main soil factors promoting *H. rhamnoides* $P_N$ are TK, TN, and SOM content; TN is the most important soil factor, and TP and AK were the limiting factors of *H. rhamnoides* $P_N$.

**Characteristics of *H. rhamnoides* fruits grown in different soil types**

The longitudinal diameter and cross diameter of the fruit and their ratio reflect the basic morphological characteristics of *H. rhamnoides* fruit. The length of the stem and the number of spines are also worth considering in the breeding process of *H. rhamnoides* as these factors impact how easy it is to pick the fruit. The weight of the fruit and the density of the fruit together determine the yield of *H. rhamnoides* (*Ling & Wu, 2007*). Table 5 shows that the longitudinal range of *H. rhamnoides* fruit from plants grown in the two soil types was 4.94 ∼8.90 mm, the diameter range was 5.42∼7.05 mm, and the stem length range was 1.28 ∼2.05 mm. Most of the fruit from *H. rhamnoides* plants in Loess soil were classified as round, while the majority of the fruit from *H. rhamnoides* plants in Aeolian soil were oval. The length of the stem of the *H. rhamnoides* fruit from Aeolian soil was greater than those from the Loess soil. The number of thorns in the plants in the Aeolian soil was relatively low, which made the fruit easy to pick, and the weight of 100 fruits was 28.28 g, which was significantly higher than the weight of 100 fruits from the Loess soil, which was 11.14 g. There were significant differences in the shape and quality of *H. rhamnoides* fruits from plants grown in the two different types of soil.

## DISCUSSION

Soil is the vehicle of plant survival and growth, driving photosynthesis by providing the necessary nutrients for the process. Understanding the gas exchange parameters and drivers of *H. rhamnoides* plants grown in different soil types is essential to evaluating the suitability of these soils for *H. rhamnoides* growth. In this study, we first compared the nutrient differences between Loess soil and Aeolian soil, and found that the SOM content

**Table 5  Analysis of *H. rhamnoides* fruit characteristics from different soil types, including morphology and quality.**

| Soil type | Longitudinal diameter/mm | Cross diameter/mm | Aspect to diameter ratio | The length of the stalk/mm | 10 cm fruit branches/piece | 10 cm number of thorns/piece | Hundred fruit heavy/g |
|---|---|---|---|---|---|---|---|
| Loessial soil | 4.84 | 5.32 | 0.91 | 1.29 | 25.11 | 2.12 | 11.24 |
| Aeolian soil | 8.90 | 7.12 | 1.25 | 2.05 | 36.00 | 0.90 | 29.30 |

of Loess soil was significantly higher than in Aeolian soil, which may be because there are more herbaceous plants under the forest and their rich microbial community promotes the production of organic matter. TK content in Aeolian soil was significantly higher than in Loess soil is related to its own weathering, while AK content was significantly lower in Aeolian soil than in Loess soil. The TP content of the two soil types were not much different, but the AP content that can be used by *H. rhamnoides* in Aeolian soil during the growing season was significantly higher than in Loess soil, which improves the ability of *H. rhamnoides* to adapt to external environmental. Under natural light conditions, the daily variation in *H. rhamnoides* $P_N$ was similar in the two soil types, with peak and trough values appearing at the same times in both soil types. However, the peak value of *H. rhamnoides* $P_N$ in Aeolian soil was significantly higher than in Loess soil. Previous studies have shown that the content of different soil elements and organic carbon content significantly affect the $P_N$ of plants (*Du et al., 2014*; *Li et al., 2020*), so we used a pathway analysis to help identify which soil factors had the largest impact. The results of the soil factor and *H. rhamnoides* $P_N$ direct diameter analysis showed that soil factors such as TN, TK, and SOM had a promoting effect on $P_N$, and their content in soil directly affected $P_N$, which is similar the findings of *Wang et al. (2019)*. TP and AK had an inhibitory effect on $P_N$, which is consistent with the research results of *Xia et al. (2014)*. The indirect diameter coefficient between soils indicated that in addition to TN, TK, and SOM directly impacting *H. rhamnoides* $P_N$, TP and AK also affect $P_N$ change, indirectly, by affecting SOM and TN content. This shows that soil factors can not only individually impact $P_N$, but the complex relationships between the soil factors can also affect $P_N$. The diurnal variation of *H. rhamnoides* leaf $T_r$ differed between the two soil types, and the $T_r$ of *H. rhamnoides* in Aeolian soil was slightly higher than in Loess soil, with $T_r$ peaking 2.5 h later in Aeolian soil than the first peak in Loess soil. Existing studies have shown that *H. rhamnoides* $T_r$ usually peaks around 12:00~13:00 (*Chen et al., 2015a*; *Chen et al., 2015b*), but can peak prematurely, which is one way *H. rhamnoides* can adapt to its external environment. There are also differences in the daily variation of *H. rhamnoides* $C_i$ between the two soil types. *H. rhamnoides* $C_i$ in Aeolian soil followed a bimodal curve, which is consistent with the change law of $P_N$ and normal physiological activity. However, daily trends in *H. rhamnoides* $C_i$ in Loess soil followed an inverted parabola, inconsistent with the change law of $P_N$. Relevant studies have shown that when $P_N$ and $C_i$ increase or decrease at the same time, the change is likely triggered by stomatal factors; but when changes in $P_N$ do not occur in tandem with changes in $C_i$ these changes are likely caused by non-stomatal factors (*Farquhar & Sharkey, 1980*; *Sharkey, 1988*; *Lawlor, 2002*; *Lawlor & Cornic, 2010*), as seen in Loess soil

in this study. Daily variation trends in $G_s$ did not differ significantly between the two soil types, but the overall *H. rhamnoides* $G_s$ of Aeolian soil was higher than Loess soil.

Overall, the gas exchange parameters of *H. rhamnoides* differed between the two soil types, supporting our first hypothesis. It is worth noting that we deliberately avoided environmental factors such as moisture, sunlight, and temperature in our study, which is the main limitation of this experiment. The gas exchange parameters of plants are related to external temperature, humidity, air pressure, and soil material composition, so in practical applications, the soil can adjusted based on the plant's own enzyme activity, luciferin changes, *etc*.

Our study on the shape and quality of *H. rhamnoides* fruits showed that the fruit of *H. rhamnoides* in Aeolian soil was oval-shaped, the fruit quality was larger than the fruit from Loess soil, and easier to pick after ripening. Compared with Chinese *H. rhamnoides* (native *H. rhamnoides*), the fruit yield of "Ulanshalin" *H. rhamnoides* is large, which can improve economic efficiency (*Wu et al., 2022*). Differences in the diurnal variation of *H. rhamnoides* gas exchange parameters between the two soil types indicate that Aeolian soil is more conducive to the growth of *Hippophae rhamnoides* ssp. *mongolica* "Ulanshalin" plants than Loess soil. *H. rhamnoides* also showed a strong adaptability to Aeolian soil, which is consistent with our second hypothesis.

## CONCLUSIONS

Our research shows that cultivating *Hippophae rhamnoides* ssp. *mongolica* "Ulanshalin" in the Yellow River Basin does not affect its high-yield potential. Aeolian soil is more conducive to the growth and development of *H. rhamnoides*, as it improves stress resistance, increases fruit yield, and creates more economic value from the plants than Loess soil. Therefore, it is recommended that Aeolian soil be used when introducing *Hippophae rhamnoides* ssp. *mongolica* "Ulanshalin" in the Yellow River Basin.

### Funding

This work was supported by the National Natural Science Foundation of China (No. 31960329), the International Science and Technology Inner Mongolia Autonomous Region Science and Technology Program Project (No. 2021GG0085, 2019GG004, 2022MS03029), and the Ordos Science and Technology Cooperation Major Special Project (2021EEDSCXQDFZ011). The funders had no role in study design, data collection and analysis, decision to publish, or preparation of the manuscript.

### Grant Disclosures

The following grant information was disclosed by the authors:
National Natural Science Foundation of China: 31960329.
International Science and Technology Inner Mongolia Autonomous Region Science and Technology Program: 2021GG0085, 2019GG004, 2022MS03029.

Ordos Science and Technology Cooperation Major Special Project: 2021EED-SCXQDFZ011.

## Competing Interests

The authors declare there are no competing interests.

## Author Contributions

- Fanjing Bu conceived and designed the experiments, authored or reviewed drafts of the article, and approved the final draft.
- Yuefeng Guo conceived and designed the experiments, performed the experiments, analyzed the data, authored or reviewed drafts of the article, and approved the final draft.
- Wei Qi performed the experiments, analyzed the data, prepared figures and/or tables, and approved the final draft.

## Data Availability

The original measurements can be found in the Supplementary File.

## Supplemental Information

Supplemental information for this article can be found online at http://dx.doi.org/10.7717/peerj.15264#supplemental-information.

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
