# Peer review of "Effects of different soil types on gas exhange parameters and fruits of Hippophae rhamnoides ssp. mongolica “Ulanshalin” plants"

_PeerJ, doi:10.7717/peerj.15264_

## Round 0.1 · original submission · Major Revisions

Dear Dr. Fanjing,

Thank you for your submission to PeerJ.

It is my opinion as the Academic Editor for your article - Effects of different types of soil on photosynthetic traits and fruits of Hippophae rhamnoides ssp. mongolica “Ulanshalin” - that it requires a number of Major Revisions.

The reviewers have invariably suggested to improve the English language of the paper for which you may take the help of colleagues or professional editors.

In addition to the reviewers comments, you have address the following concerns and resubmit the manuscript.

1. Revise the manuscript thoroughly to make it more appealing to the prospective readers, and to ensure that findings are presented and discussed in a coherent way.

2. Revise the title to make it catchy.

3. Conclusions section is missing from the abstract. You have to add the same, highlighting the novelty of the study and the future thrust.

4. Elaborate the statistical analysis part to make it more easily understandable. Give the reasons for using the path analysis.

4. It is to be ensured that references are laid out as per the journal format.

5. The quality of pictures in Figures 2 and 3 is not very good, and they need to be replaced with images of better resolution.

6. The font and letter size used by the authors need to be checked to ensure that they are as per the journal requirement.

My suggested changes and reviewer comments are shown below and on your article 'Overview' screen.

Please address these changes and resubmit. Although not a hard deadline please try to submit your revision within the next 35 days.

With regards

Anshuman Singh
Editor
Peer J Life and Environment

Reviewer 1 ·

Basic reporting

The current manuscript highlighted the role of different soil types on photosynthetic performance of Hippophae rhamnoides ssp. Although authors tried to convey the findings of the study but have few suggestions for its improvement.
The article must be improve in terms of its English and must use clear technically correct text. The introduction section needs major refinement. Kindly see the comments directly provided on ms in track change mode. Follow the journal guidelines for its improvement. In materials and methods section, authors should provide the details about photosynthetic measurement as per comments.
Figures should be appropriately described and labelled. Re-write the whole manuscript with more scientific manner.

Experimental design

Experiment was conducted in appropriate manner. Material and methods section needs some correction.

Validity of the findings

Novelty statement is missing. How these findings will be useful for future research.

Additional comments

Author re-write the whole manuscript. A major revision required before re-consideration of this manuscript.

Annotated reviews are not available for download in order to protect the identity of reviewers who chose to remain anonymous.

Reviewer 2 ·

Basic reporting

It can be seen that most of the comments mentioned by the previous reviewers have been revised by the authors. I agree with the previous reviewers, The studied contents meet the standards for this journal, the author studied the effects of two typical soils in the Yellow River Basin of China on the photosynthetic characteristics and fruits of sea-buckthorn , scientifically analyzed the experimental data, summarized the limitations in the process, and put forward some conclusions with great practical application value. I recommended it publish after minor revision.
1)The English language should be improved.
2)Clear assumptions need to be given.
3)I recommend providing some pictures of samples determined using photosynthetic instruments in the supplementary documentation.
4)The pretreatment process in Section 2.5 is very important, but it is very concise. Can it be more detailed?
5)The order of the figure numbers is incorrect. For example, Line192、201、210. Group diagrams can be listed as a, b, c, d etc.
6)Use “Fig. 1” or “Figure 1” instead of both. Remove the similar problems in your paper.
7)Some references lack the necessary issue or volume number. Complete missing data in references like issue number and volume number.
8)Figure 1 is not clear enough, and there is a big gap, which is not up to standard. Figures 2 and 3 could be larger.
9)Most of the references are older, and references from the last 5 years need to be added.

Experimental design

The experiment was well-designed. The investigation was rigorous, and the methods were described with sufficient information to satisfy the investigator's reproducibility.

Validity of the findings

The findings were validated by data and statistical analysis

Additional comments

No comment.

Reviewer 3 ·

Basic reporting

1) English is poor, lots of grammatical mistake in MS
2) Literature is quite good, but arrangements of references are not as same
3) Table and Figures are quite good
4) Hypothesis, results and Discussion must be improved.

Experimental design

Good

Validity of the findings

Good but needs to be strengthened

Additional comments

Throughout english is poor, lots of grammatical mistake in MS

Annotated reviews are not available for download in order to protect the identity of reviewers who chose to remain anonymous.

---

## Round 0.2 · accepted · Accept

Dear Dr. Fanjing
In my opinion as the Handling Editor for your manuscript 'Effects of different soil types on gas exhange parameters and fruits of Hippophae rhamnoides ssp. mongolica “Ulanshalin” plants' has been Accepted for publication.

I have assessed the revised draft, and happy to note that authors have made the necessary changes as per the reviewers' comments.